# Myco-Nanofabrication of Silver Nanoparticles by *Penicillium brasilianum* NP5 and Their Antimicrobial, Photoprotective and Anticancer Effect on MDA-MB-231 Breast Cancer Cell Line

**DOI:** 10.3390/antibiotics12030567

**Published:** 2023-03-13

**Authors:** Muthuraj Rudrappa, Raju Suresh Kumar, Shashiraj Kareyellappa Nagaraja, Halaswamy Hiremath, Pooja Vidyasagar Gunagambhire, Abdulrahman I. Almansour, Karthikeyan Perumal, Sreenivasa Nayaka

**Affiliations:** 1P.G. Department of Studies in Botany, Karnatak University, Dharwad 580003, Karnataka, India; rmuthuraj20@gmail.com (M.R.); shashiraj@kud.ac.in (S.K.N.); hmhalaswamy@kud.ac.in (H.H.); poojagambhire123@kud.ac.in (P.V.G.); 2Department of Chemistry, College of Science, King Saud University, Riyadh 11451, Saudi Arabia; sraju@ksu.edu.sa (R.S.K.); almansor@ksu.edu.sa (A.I.A.); 3Department of Chemistry and Biochemistry, The Ohio State University, 151 W. Woodruff Ave, Columbus, OH 43210, USA; perumal.11@osu.edu

**Keywords:** *Penicillium* sp., silver nanoparticles, antimicrobial activity, anticancer activity, MDA-MB-231 breast cancer cells, SPF

## Abstract

Currently, the exploration of fungal organisms for novel metabolite production and its pharmacological applications is much appreciated in the biomedical field. In the present study, the fungal strains were isolated from soil of unexplored Yellapura regions. The potent isolate NP5 was selected based on preliminary screening and identified as *Penicillium brasilianum* NP5 through morphological, microscopic, and molecular characterizations. Synthesis of silver nanoparticles from *P. brasilianum* was confirmed by the color change of the reaction mixture and UV-visible surface plasmon resonance (SPR) spectra of 420 nm. Fourier transform infrared (FTIR) analysis revealed the functional groups involved in synthesis. Atomic force microscopy (AFM) and transmission electron microscope (TEM) analysis showed aggregation of the NPs, with sizes ranged from 10 to 60 nm, an average particle size of 25.32 nm, and a polydispersity index (PDI) of 0.40. The crystalline nature and silver as the major element in NP5-AgNPs was confirmed by X-ray diffraction (XRD) and energy dispersive X-ray (EDX) analysis. The negative value −15.3 mV in Zeta potential exhibited good stability, and thermostability was recorded by thermogravimetric analysis (TGA). NP5-AgNPs showed good antimicrobial activity on selected human pathogens in a concentration-dependent manner. The MTT assay showed concentration-dependent anticancer activity with an IC_50_ of 41.93 µg/mL on the MDA-MB-231 cell line. Further, apoptotic study was carried out by flow cytometry to observe the rate of apoptosis. The calculated sun protection factor (SPF) value confirms good photoprotection capacity. From the results obtained, NP5-AgNPs can be used in the pharmaceutical field after successful in vitro clinical studies.

## 1. Introduction

Nanotechnology is a unique and innovative field of science that attracts scientists and researchers from various fields, including chemists, physicists, engineers, and biologists across the globe [1]. It is an emerging science branch that relies on the production, characterization, and applicability of small-sized particles. These nanoparticles (NPs) have applications in various fields [2]. Richard Feynman first represented the nanotechnology approach in his lecture, “There’s a plenty of room at the bottom”, at the American Institute of Technology. The word nano is derived from the Greek word ‘nanos’ meaning ‘dwarf’, and means the engineered matter exhibits one billionth in size [3]. The need of nanotechnology and nanoscience is the ability to understand, fabricate and engineer materials, devices, and systems at atomic and subatomic levels. NPs make materials lighter, stronger, faster, smaller, and more durable [4].

Nanotechnology involves the synthesis of particles having a size variation from 1–100 nm. NPs exhibit various new properties because of their large surface-to-volume ratio and small size, catalytic activity, chemical stability, high conductivity and biological properties [5,6]. Various approaches have been carried out to synthesize NPs from different metals, such as silver, gold, zinc, magnesium, copper, iron, chromium, titanium, selenium, aluminum, platinum, etc. Among these, NPs from silver have received more attention because of their specific characteristics and broad applicability in various fields, such as biosensors, drug delivery, imaging, biomedicine, and agriculture [7]. There are various methods available for the fabrication of NPs, namely the chemical method, physical method, and biological methods. Chemical and physical methods involve the use of toxic chemicals and also create hazardous by-products, which cause environmental pollution [8]. Physical methods of synthesis involve irradiation, ultra sonication, electrochemicals, and microwaves. The chemical method involves chemical reduction, condensation, the sol-gel method, and precipitation, etc. The biological nanoparticle synthesis method involves using agents such as bacteria, actinomycetes, fungi, algae, and plants [9].

The biological approach, or green fabrication of NPs, is effective, eco-friendly, more accessible, and less expensive than the chemical and physical methods [10,11]. NPs can be synthesized by extracellular and intracellular methods from fungi, plant, and microbes. Fungal extracts, plant extracts, and microbial extracts act as reducing agents, which alter the chemical nature of the metals by the mechanism of defense against toxic agents [12]. The molecules, including carbonyl groups, amines, amides, terpenoids, phenolics, proteins, pigments, and additional factors, are present in the plant extracts and microbial extracts that are involved in the synthesis of NPs [13]. Among different biological agents, fungi have been extensively used due to less cost, high biomass production, low generation of toxicity, and low energy consumption [2].

The technique for fabricating NPs from fungi is called myconanotechnology [14]. Fungi produce a large number of metabolites, enzymes, and extracellular proteins. These biomolecules act as reducing agents but are also involved in the capping of NPs; hence they are involved in controlling the size and stability of NPs [15,16]. Fungi are very effective hotspots for providing new drugs, metabolites, and antimicrobials [8]. A wide range of filamentous fungi, including *Fusarium oxysporum*, *Aspergillus* sp., *Candida albicans*, etc., are used in the development of AgNPs [17,18,19].

AgNPs have various applications in the areas of drug delivery, renewable energy technologies, luminescence, optics, electronics, textiles, cosmetics, food industry, dentistry, wound dressing, and agriculture. In addition, they have been reported for antimicrobial, antioxidant, anticancerous, antibacterial, and antiviral properties. AgNPs synthesized from fungus have diverse qualities that aid in the bio pesticide industry, improving plant growth, nutrient efficiency, synthesis of nano fertilizer, nano insecticide, nano fungicides, biosensors, etc. Recently, AgNPs attained much attention as potent anticancer drugs through triggering apoptotic activity on a number of human cancer cell lines. In the last decade, nanotechnology-based diagnosis and therapy methods for cancer treatment have attracted much attention in the medical field due to their novelty [11,20].

Breast cancer is the most common cancer in women; it is the second general cause of cancer death in women. Among the other cancers, 30% of breast cancer patients are reported every year in females. Breast cancer predominantly occurs in middle-aged to older women. Most cancers respond to chemotherapy primarily, however, further on, they develop resistance. Presently, the use of chemotherapeutic and chemo preventive agents cause undesirable side effects; hence the identification and discovery of cost-effective and biocompatible diagnostic tools for cancer is indispensable [21].

There are a number of diagnostic approaches that have been investigated to allow for the treatment of breast cancer in the early stage, such as magnetic resonance imaging, mammography, computerized tomography, positron emission tomography, ultrasound and biopsy. However, these techniques are time consuming, expensive, and unsuitable for young women. Developing a precise and quick early-stage breast cancer diagnostic method is crucial. In recent years, researchers have focused on the development of biosensors to detect breast cancer using different biomarkers. Excluding biomarkers and biosensors, microwave imaging techniques have also been intensely considered as a positive diagnostic tool for quick and cost-effective early-stage breast cancer detection [22]. Resistance to antimicrobial compounds of multidrug-resistant pathogens has been alarming in recent years and poses a huge threat to public health by causing an increase in mortality and morbidity according to the WHO. These challenges encourage researchers to identify more effective and novel antimicrobial compounds in addition to discovering novel and effective drug targeting and delivery methods. The use of nanoparticles as novel biomaterials to fully achieve this feat is currently gaining global attention. Nanoparticles could become an indispensable viable therapeutic option for treating drug-resistant infections [23]. Furthermore, most sunscreens use different chemicals to enhance their photoprotection against UV radiation, because UVA and UVB rays are capable of passing through skin cells layers and causing genetic mutations in the cells, resulting in cancer. Thus, there is an increasing demand for the use of photoprotective agents in sunscreens. Silver nanoparticles are also utilized in sunscreen preparations to enhance photoprotection and provide better and longer protection against sunburn [24].

Fungi are extensively used for the fabrication of different metal nanoparticles due to the high efficiency of the fungal metabolites in synthesis. Fungi have certain advantages that overcome other organisms, such as their economic feasibility, ease of scaling up and downstream handling, and the presence of mycelia presenting an increased surface region [25]. In the present study, the selected region for soil sample collection was not explored for fungal diversity; hence the current study focused on the isolation of novel metabolites-producing fungi from the unexplored region. To the best of our knowledge, this is the first time that the biosynthesis of AgNPs from *P. brasilianum* fungi has been reported. Considering the challenges of developing a drug against MDR microbes, the development of a drug for use against breast cancer, and to encourage the use of AgNPs in sunscreens as a photoprotective agent, the present work was conducted.

## 2. Results

From the collected soil samples, a total of 10 morphologically different fungi were isolated. Then, those fungi were sub-cultured on potato dextrose agar (PDA) media. Out of the 10 different fungi, an NP5 isolate was selected based on primary screening, which showed its good antimicrobial activity (Table 1), and thus NP5 was used for further analysis.

### 2.1. Morphological and Molecular Characterization of NP5 Fungal Strain

The selected NP5 fungal isolate was identified based on morphological and molecular characterization. The aerial mycelium was observed to be white in color with green-colored sporulation, and it produced yellowish substrate mycelia (Figure 1A,B). The SEM image of spore chain morphology exhibited how the conidia were arranged in the conidial chain, and attached to sterigmata; the sterigmata were attached to metula (Figure 1C).

The extracted genomic DNA was analyzed by PCR amplification and 18S rRNA gene sequencing. Sequencing of isolated DNA exhibited 900 base pairs by 18S rRNA gene sequencing; the sequence was submitted to the NCBI database and received accession number: OP071259. Further, the BLASTN analysis was conducted in the NCBI database to obtain the closely related organisms to construct the phylogenetic tree. The BLASTN analysis and phylogenetic tree (Figure 2) revealed 100% similarity with *Penicillium brasilianum* FUK50, *Penicillium brasilianum* FUK66, and *Penicillium* sp. CPCC480008, and they were aligned in the same branch. *Penicillium limosum* CBS339.97, *Penicillium* sp. strain R57, and *Penicillium limosum* showed 99.89% similarity. *Penicillium decumbens* isolate XXTF9, *Penicillium oxalicum* clone EF036, *Penicillium decumbens* strain ZQ001, and *Penicillium crustosum* strain 046, showed 99.78% similarity with the isolate NP5. Based on the morphological and molecular characterizations, the selected potent isolate NP5 was identified as *Penicillium brasilianum* NP5.

### 2.2. Green Synthesis of NP5-AgNPs and Their Characterization

The fabrication of AgNPs was confirmed by a change of color in the mixture of NP5 fungal extract and 1 mM AgNO_3_ from pale yellow to brown after their incubation for 24 h (Figure 3A–C). The nature of the fabricated AgNPs was further characterized by UV-visible and FT-IR spectroscopy, AFM, TEM, EDS, XRD, Zeta potential, DLS, and TGA analysis.

#### 2.2.1. UV-Visible Spectroscopy of NP5-AgNPs from *Penicillium brasilianum* NP5

The maximum absorption peak was recorded at 420 nm for the reaction mixture of AgNO_3_ and NP5 after the incubation period of 24 h. The maximum absorption peak of the NP5 fungal extract was recorded at 253 nm, and this was shifted to 420 nm after the synthesis of the AgNPs (Figure 4).

#### 2.2.2. FTIR Analysis of NP5-AgNPs

The FTIR spectrum of NP5 extract represents the major peaks at 3731, 3657, 3091, 2130, 1629, 1402, 1308, 1081, and 580, cm^−1^ (Figure 5A). The synthesized NP5-AgNPs represent the peaks at 3713, 3536, 3478, 3436, 3371, 3295, 2928, 2141, 1635, 1384, and 1075 cm^−1^ (Figure 5B). Peaks in NP5-AgNPs extract 3295, 3371, 3436, 3478, 3536, and 3713 cm^−1^ confirms the O-H stretching carboxylic acid, N-H stretching aliphatic primary amine, N-H stretching primary amine, and O-H stretching alcohol, respectively, and the peak at 2928 cm^−1^ reports the presence of C-H stretching alkane. The peaks in fungal extract 3657 to 3983 cm^−1^ correspond to O-H stretching alcohol and the peak at 3091 cm^−1^ exhibits O-H stretching carboxylic acid. The weak peaks at 2130 and 2141 cm^−1^, from both extracts, exhibit the C≡C stretching alkyne. The broad peak in extract 1629 was shifted to 1635 cm^−1^ as a sharp peak with functional group C=C stretching alkene. The sharp peaks of 1308 and 1402 cm^−1^, representing C-O stretching aromatic ester and S=O stretching sulfonyl chloride, shift to 1384 and exhibit a C-H bending aldehyde in NP5-AgNPs. The broad peaks at 580 cm^−1^ confirm the presence of C-Br stretching alkyl halides, and the sharp peak of 1081 shifted to 1075 cm^−1^, which represents the C-O stretching primary alcohol in the samples.

#### 2.2.3. AFM analysis of NP5-AgNPs

The surface topology and distribution of synthesized NP5-AgNPs were analyzed by AFM. Figure 6A represents the surface of the AgNPs showing the polydispersity and spherical shape of fabricated NP5-AgNPs. Size distribution and the histogram showed in Figure 6B,C respectively. AFM study of the fabricated NP5-AgNPs specifically helps in understanding the dissolution and agglomeration pattern of AgNPs.

#### 2.2.4. TEM analysis of NP5-AgNPs

TEM images of NP5-AgNPs showed the spherical shaped and polydispersed nature of the NPs. In addition, some of the NP5-AgNPs were agglomerated, the sizes were distributed from 10 to 60 nm, with an average particle size of 25.32 nm and a PDI of 0.40 (Figure 7A). Figure 7B shows the size distribution of NP5-AgNPs.

#### 2.2.5. EDX analysis of NP5-AgNPs

The elemental analysis of NP5-AgNPs was carried out by EDX analysis. The prominent peak was observed at 3 keV for silver metal, and this confirms the major element present in the NP5-AgNPs. The other peaks found in the EDX spectrum are the elements that act as a stabilizing and reducing agent in their synthesis from the fungal extract (Figure 8A). Approximately 67.02% of silver, 13.02% of chlorine, 11.34% of carbon, 7.67% of oxygen and 0.95% of aluminum metals were detected in the EDX analysis.

#### 2.2.6. XRD analysis of NP5-AgNPs

The crystalline nature, phase purity, and composition of synthesized NP5-AgNPs were characterized by XRD. Figure 8B shows 2Ɵ values with specific Bragg’s reflections of 111, 200, 220, and 311, which correspond to 38.1533°, 46.2409°, 64.49282° and 77.45554° Bragg’s reflections angles, respectively, in the JCPDS (card number 04-0783) database of silver. These lattice place readings confirm the face-cantered cubic crystalline nature of fabricated NP5-AgNPs.

#### 2.2.7. Zeta Potential analysis of NP5-AgNPs

The stability and surface charge potential of NP5-AgNPs in an aqueous medium was examined by Zeta potential analysis. The negative value −15.3 mV of fabricated NP5-AgNPs exhibited good stability (Figure 8C).

#### 2.2.8. TGA analysis of NP5-AgNPs

The fabricated NP5-AgNPs were subjected to TG analysis to assess their thermal stability. The thermogram of NP5-AgNPs (Figure 8D) shows a gradual decrease in the weight along with the increase in the temperature. The primary weight loss was observed between the temperatures 32.41 to 55.71 °C with 3.39% weight loss; the moisture on the surface of AgNPs may cause this weight loss. From 55.71 to 221.13 °C, the weight was stable with no loss in weight; further, 221.13 to 344.63 °C records 8.06% of weight loss, or about 0.5297 mg. The subsequent significant decrease in weight is observed between 432.73 to 459.4 °C with a 4.41% weight loss or about 0.2896 mg. The final gradual weight loss of 6.138%, about 0.4031 mg, was recorded between 741 to 849 °C. About 22.43% of the weight was lost from room temperature to 994.36 °C, and 77.57% of the weight was stable at high temperature; this suggests the thermostability of NP5-AgNPs.

### 2.3. Biological Activity of Synthesized NP5-AgNPs

#### 2.3.1. Antimicrobial Activity of NP5-AgNPs

Synthesized NP5-AgNPs with different concentrations showed good antimicrobial activity on selected pathogenic microorganisms, as shown in Figure 9A–F. The results are represented as mm of the inhibition zone in diameters. *Escherichia coli* (*E. coli*) was more susceptible to NP5-AgNPs, and showed 6.04 ± 0.94, 12.33 ± 0.76, 15.66 ± 1.06, and 19.6 ± 0.52 mm at the 25, 50, 75, and 100 µg/mL of AgNPs concentration; *Candida glabrata* (*C. glabrata*) was least susceptible to NP5-AgNPs and showed 5.56 ± 0.51, 10.26 ± 0.64, 17.66 ± 0.30 mm at 50, 75, and 100 µg/mL of AgNPs suspension, respectively. In addition, NP5-AgNPs showed a good zone of inhibition against *Candida albicans* (*C. albicans*), *Bacillus cereus* (*B. cereus*), *Shigella flexneri* (*S. flexneri*), and *Staphylococcus aureus* (*S. aureus*) at the higher concentrations in a dose-dependent manner. Thus, all the results were compared with standard streptomycin (25 µL) for bacteria and nystatin (25 µL) for *Candica* sp. The Figure 9G bar graph shows the zone of inhibition against selected pathogens.

#### 2.3.2. Anticancer Activity of NP5-AgNPs

The biosynthesized NP5-AgNPs were treated against the human breast cancer MDA-MB-231 cell line and exhibited notable proliferation inhibition of cancer cells. The morphological changes of the cancer cells after treatment revealed a variation in size, blebbing of the cell membrane, deformed cells, cell shrinkage, and roundness at varying degrees, which were observed in a concentration-dependent manner of the NP5-AgNPs, compared to the positive and negative controls showed in Figure 10A–G. The percentile cell viability decreased from 76.03 ± 1.16 to 64.913 ± 1.27, 51.84 ± 1.72, 25.97 ± 0.71, and 9.38 ± 1.50%, at the concentrations of 12.5, 25, 50, 100, and 200 µg/mL, respectively. Camptothecin (standard) exhibits 46.80 ± 0.59% cancer cell viability at the concentration of 6 µg/mL (Figure 10H). The IC_50_ of the NP5-AgNPs was found to be 41.93 µg/mL against MDA-MB-231 cancer cell line. The wells dedicated to only NPs showed a slight variation in absorbance in a concentration-dependent manner; this may be due to the presence of metabolites from fungi, which act as capping agents in the fabricated AgNPs, as compared to the MTT assay of treated cancer cells with NPs, which showed significant variation in the absorbance. The difference in the absorbance of treated cancer cells, excluding the NPs absorbance, reveals the involvement of NPs in the redox reaction with the MTT reagent to form formazan crystals (Figure 11).

#### 2.3.3. Apoptosis Assay of NP5-AgNPs by Flow Cytometry

The biofabricated NP5-AgNPs with an IC_50_ concentration of 41.93 µg/mL was considered for an AnnexinV/PI expression apoptosis study on MDA-MB-231 cancer cells. Figure 12A–C shows the percentage of cells that underwent necrosis and apoptosis, and the viable cells. The NP5-AgNPs treated cells displayed an early apoptosis of 8.3%, late apoptosis of 75.37%, necrosis of 0.14%, and 16.19% viable cells, whereas standard treated cells showed 77.52% viable cells, 19.65% apoptosis, 2.87% necrotic cells, and showed decreased cell viability when compared to the standard. Figure 12D–F shows the dead cell count of the M1 and M2 phase. Figure 12G–I exhibit the scatter plot of forward scatter (FSC) and side scatter (SSC) of untreated cells, standard treated, and NP5-AgNPs treated cells, respectively. Figure 13 exhibits the comparison of the percentile of cells that exhibited necrosis, apoptosis, and viable cells after the treatment of untreated, standard and NP5-AgNPs.

#### 2.3.4. SPF analysis of the AgNPs Synthesized from *Penicillium brasilianum* NP5

The SPF value was evaluated via the UV-visible spectrophotometric method and calculated using Equation (4). The SPF value of NP5-AgNPs at different concentrations of 0.1, 0.2, 0.3, 0.4, and 0.5 are shown in Figure 14. The highest SPF value of 14.43 ± 0.25 was recorded for the 0.5 µg/mL concentration of NP5-AgNPs, which indicates good sun protection.

## 3. Discussion

The bio fabrication of AgNPs is cost-effective and eco-friendly, with significant biological approaches in use in the field of medicine and agriculture. Fungi have been reported on for their use in the production of potent metabolites through liquid fermentation; they produce a high number of proteins and extracellular enzymes, which catalyze the heavy metal ions in the production of NPs. Fungi can grow on cheap and readily existing substrates and have the potential to generate a broad range of bioactive metabolites [26]. Yellapura was not yet explored for its diversity in fungal species; hence the present study was conducted, subsequently isolating 10 fungal strains based on primary screening. The fungal strain NP5 was further selected and identified as *Penicillium brasilianum* NP5 based on morphological and molecular characterization. Similarly, Mohammadi and Salouti isolated two *Penicillium* species from the soil for the synthesis of AgNPs [27]. Honary et al. isolated fungi identified as *Penicillium citrinum* for the synthesis of AgNPs [28], Sowmya et al. isolated *Penicillium simplicissimum* from the dumpsite of Shivamogga for the biodegradation of polyethylene [29]. In the present study, the culture supernatant from *Penicillium brasilianum* NP5 was blended with AgNO_3_ in a ratio of 1:4 with a pH of 10.5; after incubation, the color change from pale yellow to dark brown confirmed the synthesis of AgNPs. Similarly, Taha et al. demonstrated the synthesis of AgNPs based on color change [30].

The physiochemical characteristics of AgNPs are important for their biodistribution, behavior, and safety. Hence, the characterization of AgNPs is required to analyze the functional properties of synthesized NPs. The NP5-AgNPs exhibited a maximum UV absorption peak at 420 nm, which is the characteristic feature of AgNPs and confirms the synthesis of AgNPs, whereas the cell-free fungal extract exhibited a peak at 253 nm. The absorption peak of NP5-AgNPs confirms the SPR nature of AgNPs. An SPR of silver nanoparticles can be tuned throughout the near-infrared and visible region by their size and shape. Considering SPR’s applicability, an easy and controllable method for synthesizing silver nanoparticles with a defined size and shape [31]. SPR is very useful in the preliminary stages of wet chemistry synthesis, as the plasmonic response of AgNPs depends on their shape, size, dielectric environment, and upon the mutual electromagnetic interactions among particles in close proximity [32]. The absorption peak of fungal extract 253 nm is shifted to 420 nm because the metabolites present in the fungal extract were acting as capping agents during the synthesis of the AgNPs. Most of the fungal extracts exhibited the UV absorption peak between 200 nm to 300 nm, and the absorption peak at 253 nm was attributed to the aromatic amino acids of proteins that are produced by fungi [28,30]. A similar UV absorption peak of 420 nm was recorded for the synthesized AgNPs from *Penicillium verrucosum*, *Trichoderma harzianum* and *Trichoderma viride* [33,34], and 415 nm from *Penicillium italicum* [30]. The FTIR of the NP5 extract and NP5-AgNPs exhibits the chemical shift of the functional groups from the extract to the synthesis of the AgNPs. A similar chemical shift, which acts as a capping agent, was recorded by Ballottin et al. [35], with absorption peaks from the *A. tubingensis* extract and AgNPs showing similar peaks with few changes in functional groups. Similarly, Gurunathan et al. [36] also reported on the functional groups involved in the synthesis of AgNPs from *B. tequilensis* and *Calocybe indica*.

The AFM of the NP5-AgNPs exhibits a spherical, polydispersed nature, and few particles were aggregated. Similar AFM reports were given for AgNPs from *Raphanus sativus*, and *Paenibacillus* sp. [37,38]. The TEM imaging of NP5-AgNPs shows a size distribution from 10 to 60 nm, with an average size of 25 nm, spherical shape, and a PDI of 0.40, which indicates the polydispersity of the fabricated NPs. PDI is used to determine the average uniformity of the particles within the solution, which ranges from 0 to 1. A PDI value of more than 0.1 reports a polydispersed nature; less than 0.1 confirms a monodispersed nature [39]. A similar polydispersity and size distribution has been exhibited by AgNPs from *Alternaria* sp. and *Aspergillus sydowii,* with particle sizes ranged from 1–21 nm [40,41].

The elemental analysis of NP5-AgNPs exhibited a prominent silver peak at 3 keV, indicating that silver is the primary element in the fabricated AgNPs. The weaker peaks of carbon, oxygen, potassium, silicon and other recorded elements in the nanoparticles are due to fungal bioactive metabolites, such as proteins and enzymes, which cap and impart the stability of AgNPs [42]. Pallavi et al. [43] also found that biosynthesized AgNPs from the *Streptomyces hirsutus* strain SNPGA-8, exhibit prominent silver peak at 3 keV with 22.24% silver in the synthesized NPs along with other trace elements. NP5-AgNPs exhibit the four major diffraction peaks of silver observed at Bragg’s planes (111, 200, 220, 311) and these lattice place values represent the face-centered cubic nature of the NPs [43]. The obtained peaks indicate that the Ag^+^ is completely reduced to Ag^0^ by stabilization and the reduction of the NP5 extract; further, the other peaks, along with the silver peaks, represent the proteins and other bioorganic compounds in the extract [44]. Similar XRD patterns were reported by Ma et al. [45], where AgNPs from *Penicillium aculeatum* Su1 showed lattice plane values at an angle of 38.30°, 44.15°, 64.59°, and 77.57°, respectively.

The Zeta potential analysis indicates the dispersion, and surface charge of NPs present in the medium. The Zeta potential value for NP5-AgNPs displayed a sharp peak at −15.3 mV, the negative value indicates the relative stability of AgNPs. The incipient stability is exhibited between ≤−30 mV to ≥+30 mV, thus the NPs with less than −30 mV are considered an ionic [46].

The TGA of NP5-AgNPs showed good thermostability and 77.57% of the weight was retained even at 990 °C; this shows that the NP5-AgNPs with stand high temperatures. Similar thermostability was recorded for AgNPs from *Asphodelus aestivus* [47], and *Galphimia glauca* [48].

The green fabricated NP5-AgNPs exhibited concentration-dependent antimicrobial activity. In our study, *E. coli* is more susceptible to NP5-AgNPs, and *C. glabrata* showed resistance compared to the other microbial pathogens used. Many studies elucidated the probable modes of action for the antimicrobial mechanism shown by AgNPs, namely, the interaction between silver NPs and a bacterial cell cause cellular membrane damage, ribosome disassembly, enzyme inactivation, protein denaturation, the production of reactive oxygen species, or disruption of the electron transport chain, and cell death occurs in the pathogenic microbe. Similar results were reported by Swamy et al. [49], where NPs from *Amycolatopsis* sp. strain MN235945 showed greater activity against *E. coli*, then other pathogens in a concentration-dependent manner. AgNPs from *Artocarpus lakoocha* fruit extract exhibited significant antimicrobial activity against *S. pneumonia* and was less sensitive against *S. aureus* [50]. *Galphimia glauca* leaf-mediated AgNPs also showed substantial antimicrobial activity against most of the microbial pathogens, whereas GG-NPs exhibited potent activity against *P. aeruginosa* and *C. glabrata* [48]. This significant effect of AgNPs against microbial pathogens is due to the small size and electrostatic force of the attraction between the less negative to positive charge of the AgNPs and the negatively charged microbial cell membrane; this attraction converts the cells chemical and physiological properties, which break the normal physiological properties including the permeability, respiration, etc., of cells [51,52]. The attachment of NPs to lipopolysaccharides in the cell membrane disrupts the cell wall and helps with the entry of the NPs into the cell, and the binding of cellular components, such as proteins, enzymes and DNA, to the NPs causes cell death [49]. AgNPs have also been reported for on regarding the generation of reactive oxygen and nitrogen species, where they create oxidative stress upon DNA, and several other cell constituents, and disturb the main functions of bacterial cells [52].

The fabricated NP5-AgNPs exhibit significant anticancer activity on the MDA-MB-231 human breast cancer cell line in a concentration-dependent manner with an IC_50_ of 41.93 µg/mL. During the MTT assay, the yellow colored MTT reagent reacted with mitochondrial enzyme succinate dehydrogenase, which is released from cancer cells after the cell’s death, resulting in the formation of bluish-purple formazan crystals. Since the MTT assay is a colorimetric assessment, the results were recorded in absorbance at 570 nm, based on the color intensity, the cell viability was recorded and compared to the control [53]. Similar results were reported by Gurunathan et al. [36], who biosynthesized AgNPs from *Bacillus tequilensis* and *Calocybe indica* that inhibited MDA-MB-23 cell proliferation with an IC_50_ of 10 μg/mL and 2 μg/mL, respectively. In addition, AgNPs from *Bacillus funiculus* showed a dose-dependent decrease in the MDA-MB-23 cell with an IC_50_ of 8.7 µg/mL [21]. Edetsberger et al. [54] reported that NPs ≤20 nm in size could penetrate cells without considerable endocytosis and were distributed in the cytoplasm, causing mitochondrial DNA damage and cell death. NPs with ≤20 nm have a greater cellular uptake than that of AgNPs ≤100 nm in human glioma U251 cells [55]. In this case, AgNPs from the NP5 extract exhibit an average particle size of 25 nm and have the capacity to decrease the cell viability of cancer cells. Sangour et al. [56] clearly demonstrated that treatment with AgNPs did not affect normal cells and did not show any phenotypic changes via Acridine orange staining, whereas MCF-7 breast cancer cells treated with AgNPs exhibited phenotypic changes such as vacuole degeneration, apoptosis, a change in shape of some cells to rod shape, and necrotic cells. Similarly, Balkrishna et al. [57] reported that the lowest concentration of AgNO_3_ can also cause damage to normal cells, whereas AgNPs from *Putranjiva roxburghii* seed extract exhibited no toxicity at the same concentration. Rajiri et al. reported that *Delonix regia* extract and AgNPs both had few effects on normal cells and did not affect the cell proliferation of normal cells; whereas, in the case of MCF-7 breast cancer cells, the combination of *D. regia* extract and AgNPs exhibited a significant inhibition of cancer cells [58]. Several studies have been conducted that evaluated cytotoxicity without the use of interference controls. Synthesized NP5-AgNPs alone, exhibited a slight variation in absorbance in an MTT assay, due to interference in the redox reaction of MTT salts [59]. Most redox-active metals can catalyze the reduction of tetrazolium salts, and AgNPs are also capable of catalyzing redox reactions [60]. Coating and size altered the magnitude of the reaction kinetics, suggesting that various AgNPs can cause specific levels of interference [59].

The verification of apoptosis was performed by the Annexin V-FITC/PI apoptosis method. The NP5-AgNPs with an IC_50_ of 41.93 µg/mL showed good apoptosis induction against the MDA-MB-231 cell line. The initiation of the apoptotic signaling pathways to activate cancer cell death is the main action of many anticancer drugs [39]. Treatment with AgNPs can generate reduced cell viability, changes in cell shape, and increased lactate dehydrogenase (LDH) release, which leads to cell necrosis and apoptosis [30]. Similarly, the treatment of albumin-coated AgNPs with an IC_50_ of 5 µM showed good anticancer activity and different stages of cell death [61]. Adeyemi and Otohinoyi [62] also conducted flow cytometry to validate the difference in organic Ag, Au, Ag/Au NPs treatment on the MDA-MB-231 cell line and found that AgNPs and AuNPs have strong apoptotic activity, whereas the Ag/AuNPs showed no apoptotic activity upon the cells.

Hazardous genetic mutations, such as pyrimidine dimer formation, inflammation, and skin cancer, are induced by UV-B radiation. The UV-B radiation activates and enhances the formation of ROS and free radicals in the skin, induces loss and destructive damage in cellular function, and also regulates the oxidative stress of skin macromolecules [63]. Compounds with an SPF value of 6–10 provide lower protection, 15–25 moderate protection, 30–50 provides high protection, and more than 50+ SPF compounds provide very high UV protection [64]. SPF compounds have the potency to absorb UV light to protect cells from photodamage. Similarly, Salunkhe et al. [24] reported on the enhancement of SPF in commercially available sunscreens with the addition of the pigment Xanthomonadin extracted from *Xanthomonas* sp. and the AgNPs synthesized from Xanthomonadin; they observed that the SPF increased with the addition of the pigment and AgNPs compared to the SPF specified in the sunscreens. In our study, the AgNPs from NP5-AgNPs showed photoprotective activity by absorbing UV radiation with moderate SPF values; in future, it can be used to enhance the SPF of sunscreens.

In summary, the isolation of a potent fungal isolate from soil was carried out, and this was identified as *Penicillium brasilianum* NP5 based on morphological, microscopic, and molecular characterization. Further, the biofabrication of AgNPs was achieved by mixing AgNO_3_ with the NP5 fungal extract, which acted as a capping agent during synthesis. The synthesized NPs were characterized by UV-visible spectroscopy, FTIR, AFM, TEM, EDX, XRD, Zeta potential, and TG analysis, which confirmed the nature of the AgNPs. Biofabricated NP5-AgNPs exhibited potent antimicrobial activity against human pathogens, significant anticancer activity against the MDA-MB-231 breast cancer cell line, with confirmation by flow cytometric analysis, and good SPF value for photoprotection.

## 4. Materials and Methods

### 4.1. Chemicals and Reagents

Potato dextrose broth (HiMedia, Mumbai, India), agar, beef extract, peptone, sodium chloride (NaCl), silver nitrate (AgNO_3_), fetal bovine serum (#RM10432, HiMedia), Dulbecco’s Modified Eagle Medium (DMEM) high glucose (#AL1111, HiMedia, Bangalore, India), streptomycin, penicillin, amphotericin-B, camptothecin (#C9911, Sigma, Bangalore, India), potassium bromide (KBr), dimethyl sulfoxide (DMSO) (#PHR1309), trypsin-EDTA, PBS, FITC-AnnexinV (Cat No: 51-65874X, BD Biosciences, Bangalore, India), and propidium iodide (PI).

### 4.2. Collection and Isolation of Fungi from Soil Sample

A total of 8 soil samples were collected from different forest regions of Yellapur, (latitude 14°51′17.3″ N and longitude 74°44′49.3″ E), Uttara Kannada, Karnataka, India. Samples were collected from 3–4 cm deep using a sterile spatula, stored in sterile polythene bags, and brought to the microbiology research laboratory, Department of Botany, Karnatak University Dharwad, Karnataka, India, and stored for further processing. Isolation of fungi from soil was performed by standard serial dilution method from 10^−1^ to 10^−5^ on PDA. The sample was spread uniformly with the help of an L-shaped glass rod. The plates were kept for incubation at 28 ± 2 °C for 3–5 days. After the incubation, the plates were examined for fungal growth and the different fungal isolates were sub cultured, pure cultured, and maintained in PDA media for further analysis.

### 4.3. Primary Screening of Fungi Isolates

Primary screening of fungal isolates for antimicrobial potency was carried out by the agar well diffusion method. The isolated fungus isolates were grown in PDA broth, and the cell-free extract was used as a fungal extract. The antimicrobial activity was evaluated by using a nutrient agar medium. The agar plate surface was seeded with 100 µL of microbial inoculum (*E. coli* (MTCC40), *S. aureus* (MTCC6908), *C. albicans* (MTCC227) over the entire medium. The wells were made on the medium using a cork borer, and 50 µL of the fungal extract was introduced into the respective wells. The agar plate was incubated at 30 °C overnight; after 12 h of incubation, the plates were examined for the formation of a zone around the wells, which confirms the inhibitory action of fungi metabolites on the growth of pathogens.

### 4.4. Morphological and Molecular Characterization of NP5 Isolate

On the basis of growth intensity, color of substrate, and aerial mycelia on PDA media, assessment of the morphological features of the NP5 isolate was conducted. Molecular identification of the potent isolate was carried out by gene sequencing. Freshly cultured potent isolate chromosomal DNA was isolated by using a spin column kit (HiMedia, India). The fungal 18S rRNA gene was amplified using polymerase chain reaction and purified by Exonuclease I–Shrimp Alkaline Phosphatase (Exo-SAP). Then, sequencing was conducted through the Sanger sequencing method in an ABI 3500xl genetic analyzer. The recorded sequence was submitted to the National Centre for Biotechnology Information (NCBI) database to obtain the accession number by BLAST analysis, to find the closest culture sequence that identifies regions of local similarity between sequences. The DNA sequences were aligned and an evolutionary tree was designed by the neighbor-joining method using MEGA 7.0 software [65].

### 4.5. Preparation of Fungal Extract for Nanoparticle Synthesis

The NP5 isolate was inoculated into 200 mL PDB broth in a conical flask and incubated at 30 °C for 3–5 days. After incubation, broth containing mycelium was filtered, centrifuged and the cell-free supernatant was used as a fungal extract.

### 4.6. Synthesis of AgNPs Using NP5 Fungal Extract

Approximately 1 mM of AgNO_3_ solution was blended with cell-free supernatant in the ratio 4:1, and the pH was balanced to 10.5, then it was kept for incubation in the dark for 24 h. The reduction of the silver ions was observed by the color change of the mixture from pale yellow to brown. The synthesized AgNPs were collected by centrifuging at 8000 rpm for 30 min and the pellet of NPs were successfully washed with distilled water and stored to carry out further analysis [66].

### 4.7. Characterization of Synthesized AgNPs Synthesized from NP5 Fungal Extract

Characterizations were conducted through various analytical methods, including UV-Visible spectroscopy, FTIR, AFM, TEM, EDX, XRD, Zeta potential, and TGA.

#### 4.7.1. UV-Visible Spectroscopy Analysis of Synthesized NP5-AgNPs

The bioreduction of AgNPs was monitored by the color change of the reaction mixture from pale yellow to brown. Further, it was confirmed by the maximum absorption peaks of the solution using a UV 9600A UV-Visible spectrophotometer (Shanghai Metash Instruments Co., Ltd., Shanghai, China) at wavelengths scanning from 200–700 nm in which distilled water was used as a blank.

#### 4.7.2. Fourier Transform Infrared Spectroscopy of Synthesized NP5-AgNPs

The functional groups present in the NP5 fungal extract and the NP5-AgNPs were identified using FTIR analysis. The samples were dried in thermostatted desiccators to avoid water molecules. Then, the samples were mixed with potassium bromide crushed to a fine powder, and a pellet was prepared using a hydraulic press. The prepared pellets were analyzed using a NICOLET 6700 FTIR Spectrophotometer (Waltham, MA, USA) and recorded using the transmittance method between 500 cm^−1^ and 4000 cm^−1^ at a resolution of 4 cm^−1^.

#### 4.7.3. Atomic Force Microscopic Analysis of Synthesized NP5-AgNPs

Atomic force microscopy of fabricated AgNPs was performed to study the morphology and distribution of the particles. The suspension of AgNPs was prepared in distilled water and ultrasonicated for 5 min. A very thin film of NP5-AgNPs suspension was prepared on clean glass slides and allowed to dry. Then, the slides were analyzed on an oscillated cantilever attachment using a Nanosurf Flex AFM (Liestal, Switzerland) instrument.

#### 4.7.4. Transmission Electron Microscopy of NP5-AgNPs

The shape, size, and surface morphology of the synthesized AgNPs were determined using a TEM instrument (Hitachi, Model: S-3400N, California, CA, USA) with an accelerating voltage of 80 kV. A single drop of the suspension was placed on the surface of the copper grid, and the images were recorded up to the magnification of 6000× to 8000×. The size, shape, and surface morphology of the AgNPs were recorded using ImageJ 1.45 s software. The polydispersity index was recorded by calculating the average radius of the NPs and the standard deviation with the help of equation
p = σ/R_Avg_(1)
where σ = standard deviation of a radius of a batch of nanoparticles, p = dispersity, and RAvg = average radius of nanoparticles.

#### 4.7.5. Energy Dispersive X-ray Analysis of Synthesized NP5-AgNPs

The AgNPs were placed on carbon tape, which was fixed to a stub, followed by gold plating by sputtering with gold. The gold plated sample was examined to determine the elements present in the AgNPs sample by elemental analysis using an EDX (JEOL, JSM-IT 500LA, Tokyo, Japan) instrument.

#### 4.7.6. X-ray Diffractometric Analysis of NP5-AgNPs

The identification of the crystalline nature and particle size of fabricated AgNPs was characterized by using a Rigaku Miniflex 600 (Austin, TX, USA) instrument for the X-ray diffractometry. The AgNPs were placed on the cavity slide and a smooth surface was made by pressing. The XRD spectra were analyzed between 30° and 80° by X-ray diffractometer with a CuKα radiation filter (λ = 0.15418 nm) running at 40 kV and 30 mA, 2Ɵ/Ɵ scanning mode. The sizes of the AgNPs were determined with the help of Debye–Scherrer’s equation
D = 0.89λ/βcosƟ(2)where D is the particle size (nm), λ is the X-ray wavelength, β is the full line width at half maximum (FWHM) elevation of the important peak and Ɵ is the refractive (Bragg’s) angle.

#### 4.7.7. Zeta Potential Analysis of Synthesized NP5-AgNPs

To examine the surface charge and stability of the AgNPs, Zeta potential analysis was performed. The AgNPs suspension was prepared by dissolving the AgNPs in distilled water followed by ultrasonication and centrifugation. The saturated suspension was taken in the cuvette and analyzed with a dispersion medium viscosity of 0.894 mPa.s, a conductivity of 0.275 mS/cm, and an electrode voltage of 3.3 V using a Horiba Scientific Nanoparticle Analyzer (SZ-100, Kyoto, Japan).

#### 4.7.8. Thermo Gravimetric Analysis of Synthesized NP5-AgNPs

The thermal behavior of fabricated NP5-AgNPs with respect to their temperature and weight was studied by TGA analysis. In total, 5 mg of the AgNPs sample was weighed and placed inside the furnace. The temperature was gradually increased by passing inert gas and with correspondence to the temperature weight, the sample was measured. A TGA was carried out with an increase in heating rate of 10 °C/min from room temperature to 1000 °C using a TA instrument (SDT Q600 and DSC Q20, New castle, DE, USA).

### 4.8. Biological Activity of Synthesized NP5-AgNPs

#### 4.8.1. Antimicrobial Activity of Synthesized NP5-AgNPs

The antimicrobial capability of fabricated AgNPs was analyzed on different pathogenic microorganisms using the agar well diffusion method. The Gram-negative bacteria *E. coli* (MTCC40), *S. flexneri* (MTCC1457), Gram-positive bacteria *S. aureus* (MTCC6908), *B. cereus* (MTCC11778), and fungal strains *C. albicans* (MTCC227), *C. glabrata* (MTCC3019) were collected from MTCC, Pune, India. One mg/mL AgNPs suspension was prepared in sterilized distilled water. Pathogenic microbes were grown overnight in Muller Hinton broth (HiMedia, Mumbai, India). The antimicrobial activity was conducted on nutrient agar medium; about 0.5 Mc-Farland concentrations of pathogenic organisms were swabbed onto surface nutrient agar in plates and 6 mm wells were made using a sterilized cork borer. Different volumes from the 1 mg/mL concentrated solution of AgNPs suspensions (25, 50, 75, and 100 µL) were transferred to the wells, and 25 µL of streptomycin was used as a positive control for bacteria and nystatin for *Candida* sp., while sterilized distilled water was taken as a negative control. All the plates were incubated at 37 °C for 24 h; then, after the incubation period, the zone of inhibition was recorded [67,68]. The experiments were performed in triplicate.

#### 4.8.2. Anticancer Activity of Synthesized NP5-AgNPs

The anticancer activity of NP5-AgNPs on MDA-MB-231 cancer cell lines (Human Breast Cancer) was carried out using an MTT assay according to the protocol of Rudrappa et al. [39]. The MDA-MB-231 human breast cancer cell line was procured from NCCS, Pune, India, and cultured in DMEM medium to which 10% fetal bovine serum (FBS, #RM10432, HiMedia, Mumbai, India), streptomycin (100 μg/mL), penicillin (100 IU/mL), and amphotericin-B (5 μg/mL) were added before being incubated in a 5% CO_2_ humidified incubator at 37 °C until confluence occurred. After successful proliferation, 200 µL of cell suspension was added in a 96-well plate with a density of 20,000 cells/well, followed by incubation in a 5% CO_2_ incubator at 37 °C for 24 h. Different concentrations of NP5-AgNPs suspensions (12.5, 25, 50, 100, and 200 µg/mL) were supplemented to the culture medium in 96-well plates and camptothecin (6 µg/mL #C9911, Sigma) was taken as a positive control; cells without any compound were used as a negative control. The plate was incubated at 37 °C for 24 h within a 5% CO_2_ atmosphere, then 50 µL of MTT reagent was added to make up the final volume of 0.5 mg/mL. The plate was covered with aluminum foil to avoid exposure to light and incubated for 3 h. A total of 100 µL dimethyl sulfoxide (DMSO) solution was added to every well to dissolve the formazan crystals on the last step. The absorbance at 570 nm was recorded by the ELISA reader, and the IC_50_ value was determined using a linear regression equation. In addition, the MTT assay was conducted in a way where by the wells dedicated to containing only NPs of different concentrations were used in MTT analysis, excluding cancer cells. The experiments were performed in triplicate and the cell viability was evaluated using Equation (3) [69] as follows:Cell viability % = sample absorbance/control absorbance × 100(3)

#### 4.8.3. Apoptosis Assay of NP5-AgNPs by Flow Cytometry

The rate of cell death in the MDA-MB-231 cancer cell line after being treated with an IC_50_ concentration of NP5-AgNPs was examined by flow cytometry. Fresh cultured MDA-MB-231 cancer cells were added in a 96-well plate with a density of 0.5 × 10^6^ cells/2 mL and incubated at 37 °C for 24 h. Further cells were treated with an IC_50_ concentration of biosynthesized NP5-AgNPs suspension; culture medium without any treatment was used as a control. IC_50_ of the camptothecin (6 µg/mL) was used as a standard control. After the treatment, the nursed cells were washed two times with PBS, and the PBS was then removed completely, followed by the addition of 200 µL of trypsin-EDTA solution and incubated for 3–4 min at 37 °C; later, 2 mL of culture medium was supplemented. Then, the cells were transferred directly into 12 × 75 mm polystyrene tubes and centrifuged at 300 rpm for 5 min at 25 °C. The pellet was collected and washed with PBS. Then, 5 µL of Fluorescein Isothiocyanate-Annexin V (FITC-Annexin V) was mixed with the pellet with a gently vortexed motion and incubated at 25 °C for 15 min in the dark, followed by the addition of 5 µL of Propidium Iodide (PI) and 400 µL of 1X binding buffer and analyzed by flow cytometry [50,70].

#### 4.8.4. Determination of SPF Value of Synthesized NP5-AgNPs

The different concentrations (0.1, 0.2, 0.3, 0.4, and 0.5 µg/mL) of NP5-AgNPs samples were prepared by dissolving NP5-AgNPs in distilled water. The absorption spectrum of each sample was measured using a UV-visible spectrophotometer between the wavelengths of 290–320 nm at an interval of 5 nm. The SPF value was calculated by using the Mansur Equation (4).
(4)$$SPF=CF\times \sum_{290}^{320}EE\lambda \times I\left[\lambda \right]\times Abs\left[\lambda \right]$$
where, CF—Correction factor (10), EE—Erythema effect spectrum at wavelength λ, Abs—Absorbance of the sample, the value of EE λ × I [λ] is constant and it is 1. The summation of all wavelengths of SPF gives the SPF of the compound [71,72]. The experiments were performed in triplicate.

### 4.9. Statistical Analysis

Statistical analysis was carried out using Origin Pro 10.0.0.154 (2023) and SPSS statistics 2020 software. The results of the antimicrobial, anticancer, and SPF experiments are expressed as the means ± standard deviation (SD, for each group *n* = 3) to produce graph preparations. The data were subjected to an analysis of variance (ANOVA) using Origin Pro (version 10.0.0.154) and SPSS statistics 2020 software. The comparison between the treatments was analyzed using one-way ANOVA at a significant level of *p* ≤ 0.01 with Tukey’s B test.

## 5. Conclusions

The green approach is eco-friendly and cost-effective for the synthesis of NPs. In the current study, among the isolated fungi, the NP5 strain exhibited significant antimicrobial activity and was identified as *Penicillium brasilianum* NP5 based on morphological, microscopic, and molecular characterization. The AgNPs were synthesized by using NP5 fungal metabolites and confirmed by a change in the color and SPR at 420 nm by UV-visible spectrum. FTIR results showed the broad range of bioactive functional groups involved in AgNPs synthesis. The spherical shape, size, agglomeration, and polydispersity nature of the NP5-AgNPs were determined by AFM and TEM analysis. Significant stability in the face-centered cubic structure was observed in the XRD, Zeta and TG analyses. Biofabricated AgNPs exhibited good antimicrobial activity against all of the selected pathogenic microbes. The cell proliferation of the MDA-MB-231 breast cancer cell line was significantly reduced by NP5-AgNPs with an IC_50_ of 41.93 µg/mL; further, an apoptosis study revealed comparatively good enhancement of apoptosis in breast cancer cells. The calculated SPF was found to be 15 against UV radiation at the lower concentration, which reveals the photoprotective nature of NP5-AgNPs. In conclusion, from the obtained results, NP5-AgNPs demonstrate the use of NPs in various biomedical fields as antimicrobial, anticancer, and photoprotection agents.

## Figures and Tables

**Figure 1 antibiotics-12-00567-f001:**
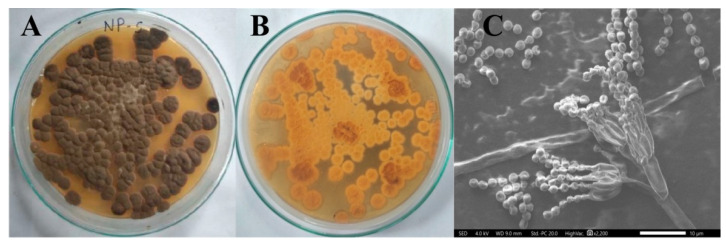
Morphology of *Penicillium brasilianum* NP5; (**A**) Aerial mycelium (1× magnification), (**B**) Substrate mycelium (1× magnification), (**C**) SEM image (10 µm).

**Figure 2 antibiotics-12-00567-f002:**
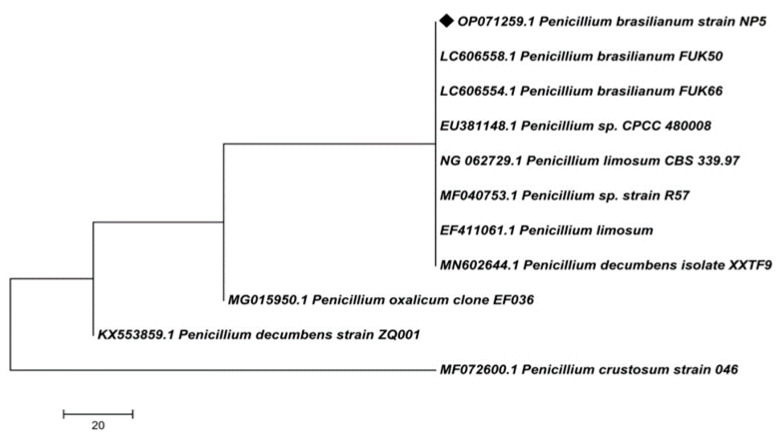
Phylogenetic tree of *Penicillium brasilianum* NP5.

**Figure 3 antibiotics-12-00567-f003:**
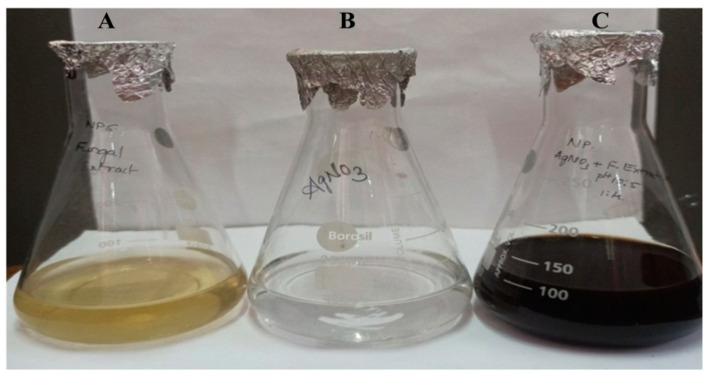
Visual observation of the biosynthesis of AgNPs; (**A**) *Penicillium brasilianum* NP5 extract, (**B**) AgNO_3_ solution, and (**C**) Synthesis of silver nanoparticles.

**Figure 4 antibiotics-12-00567-f004:**
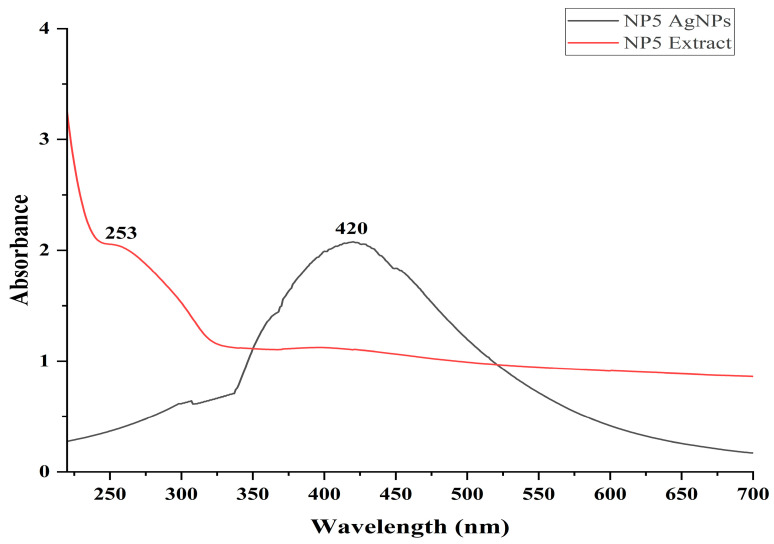
UV-Visible absorption spectrums of *Penicillium brasilianum* NP5 extract and synthesized AgNPs from *Penicillium brasilianum* NP5.

**Figure 5 antibiotics-12-00567-f005:**
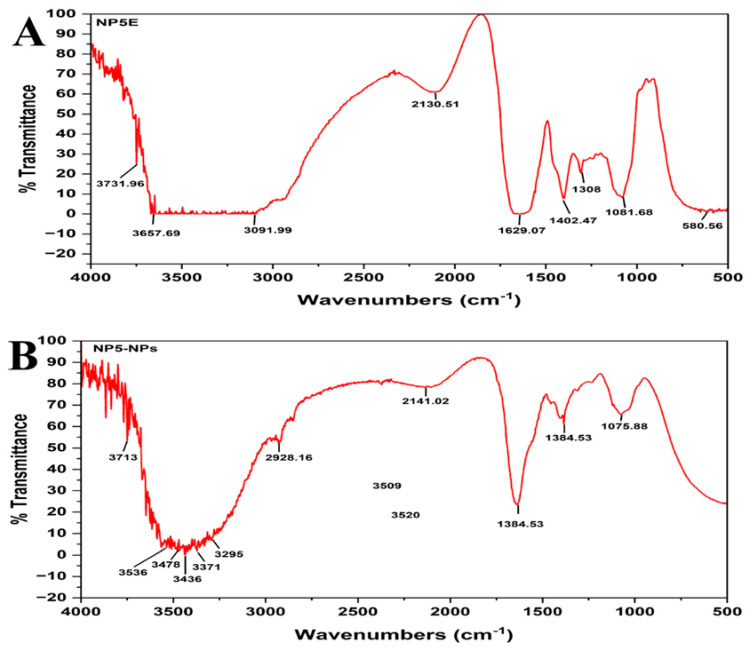
FT-IR analysis; (**A**) *Penicillium brasilianum* NP5 Extract, (**B**) Synthesized AgNPs from *Penicillium brasilianum* NP5.

**Figure 6 antibiotics-12-00567-f006:**
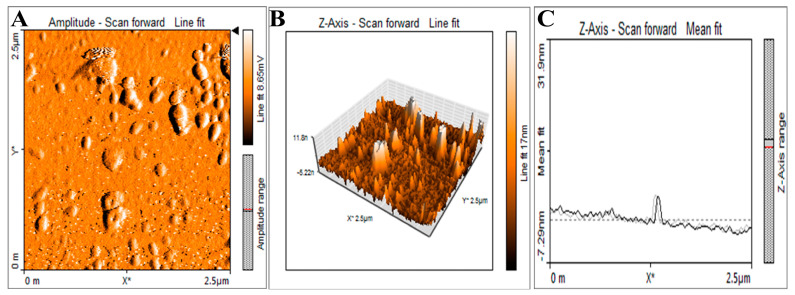
AFM images of synthesized AgNPs from *Penicillium brasilianum* NP5; (**A**) 2D image, (**B**) Size distribution, and (**C**) Histogram showing distribution of NP5-AgNPs.

**Figure 7 antibiotics-12-00567-f007:**
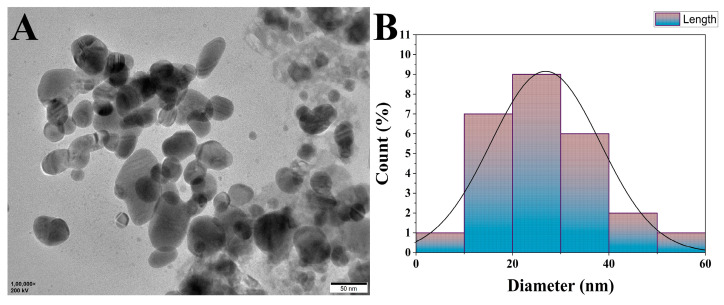
HR-TEM image of synthesized AgNPs by *Penicillium brasilianum* NP5; (**A**) Agglomeration of silver nanoparticles, (**B**) Histogram showing size distribution.

**Figure 8 antibiotics-12-00567-f008:**
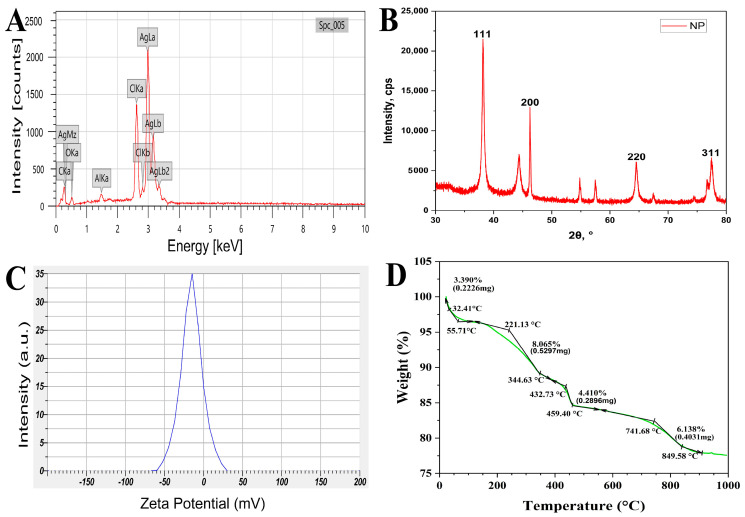
(**A**) EDX analysis, (**B**) XRD analysis, (**C**) Zeta potential, (**D**) TGA thermogram of the synthesized AgNPs from *Penicillium brasilianum* NP5.

**Figure 9 antibiotics-12-00567-f009:**
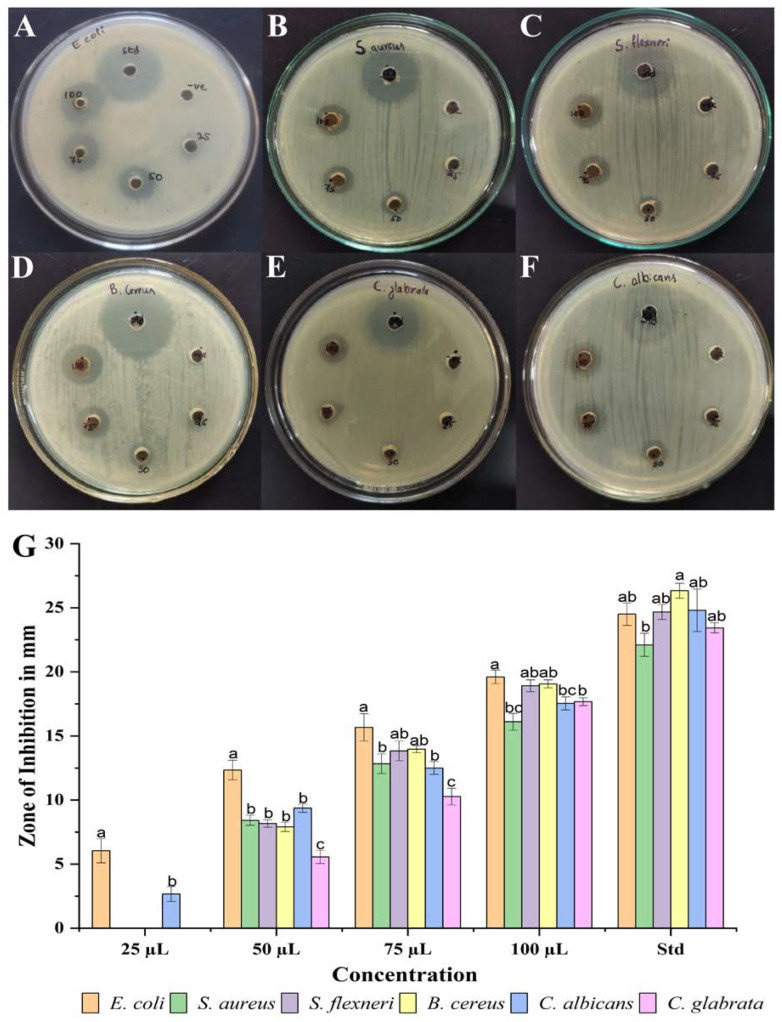
Antimicrobial activity of synthesized AgNPs from *Penicillium brasilianum* NP5 against pathogenic microorganisms; (**A**) *E. coli*, (**B**) *S. aureus*, (**C**) *S. flexneri*, (**D**) *B. cereus*, (**E**) *C. glabrata*, (**F**) *C. albicans*, and (**G**) graph showing the zone of inhibition against selected pathogens. Data are statistically significant at *p* ≤ 0.01 by one-way ANOVA with Tukey’s B test; error bars are based on three independent experiments (means ± SD, *n* = 3). Different letters at the same concentration refer to data that are significantly different.

**Figure 10 antibiotics-12-00567-f010:**
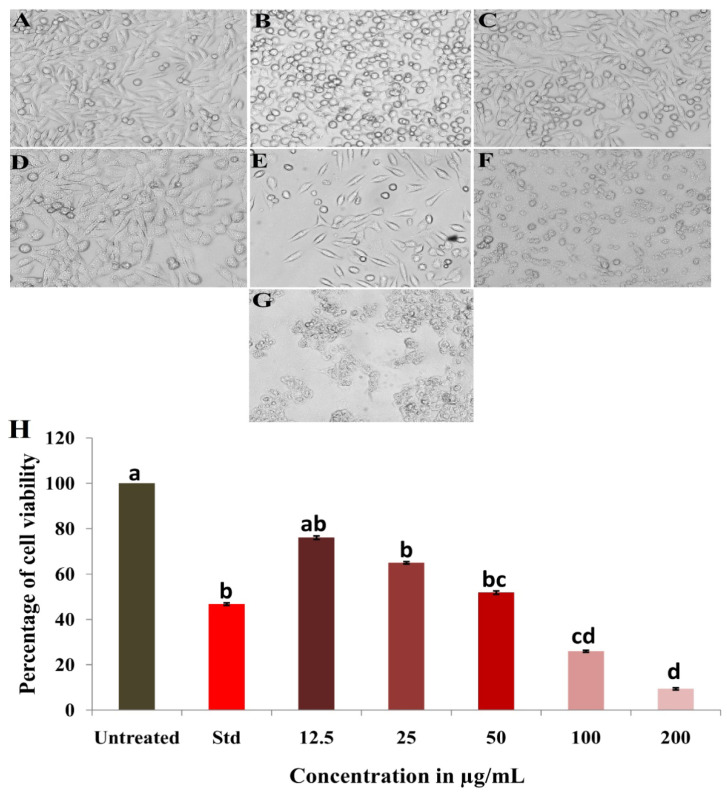
Morphological changes in the MDA-MB-231 cell line after exposure to different concentrations of AgNPs synthesized from *Penicillium brasilianum* NP5; (**A**) Untreated, (**B**) Standard, (**C**) 12.5 μg/mL, (**D**) 25 μg/mL, (**E**) 50 μg/mL, (**F**) 100 μg/mL, (**G**) 200 μg/mL of NP5-AgNPs, (**H**) Graph showing the percentage of cell viability on MDA-MB-231 cancer cells at different concentrations of NP5-AgNPs. Data are statistically significant at *p* ≤ 0.01 by one-way ANOVA with Tukey’s B test, error bars are based on three independent experiments (means ± SD, *n* = 3). Different letters at the concentrations refer to the data that are significantly different.

**Figure 11 antibiotics-12-00567-f011:**
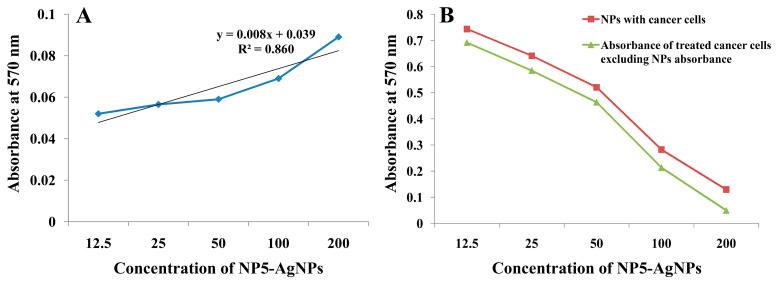
UV absorbance of MTT assay; (**A**) Only NP5-NPs, (**B**) NP5-AgNPs with cancer cells, and the difference in the absorbance of treated cancer cells, excluding NPs absorbance.

**Figure 12 antibiotics-12-00567-f012:**
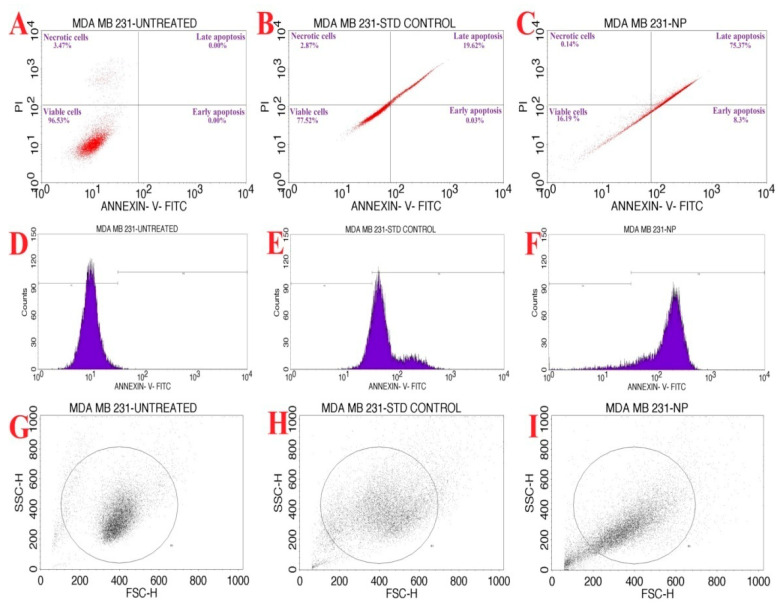
Flow cytometric analysis of synthesized AgNPs from *Penicillium brasilianum* NP5 against the MDA-MB-231 cancer cell line. Quadrangular plot of Annexin V/PI expression on MDA-MB-231 cells; (**A**) Untreated, (**B**) Standard, (**C**) NP5-AgNPs treated cells. Cells count on Annexin-V/FITC; (**D**) Untreated, (**E**) Standard, (**F**) NP5-AgNPs treated cells. Scatter plot of forward scatter (FSC) against side scatter (SSC); (**G**) Untreated, (**H**) Standard, (**I**) NP5-AgNPs treated cells.

**Figure 13 antibiotics-12-00567-f013:**
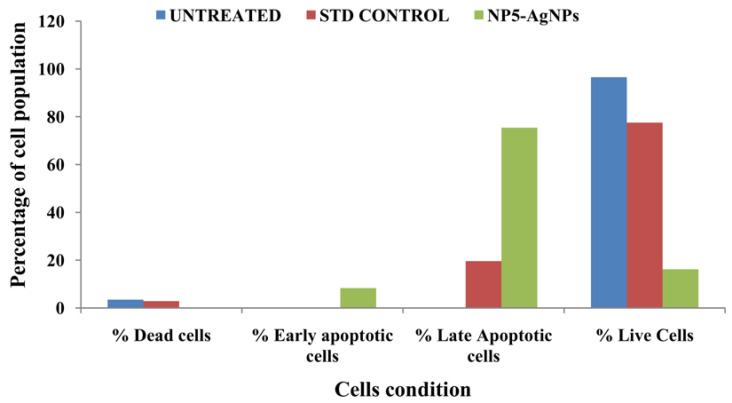
Graph showing the percentage of live, apoptotic and necrotic cells after the treatment of synthesized AgNPs from *Penicillium brasilianum* NP5.

**Figure 14 antibiotics-12-00567-f014:**
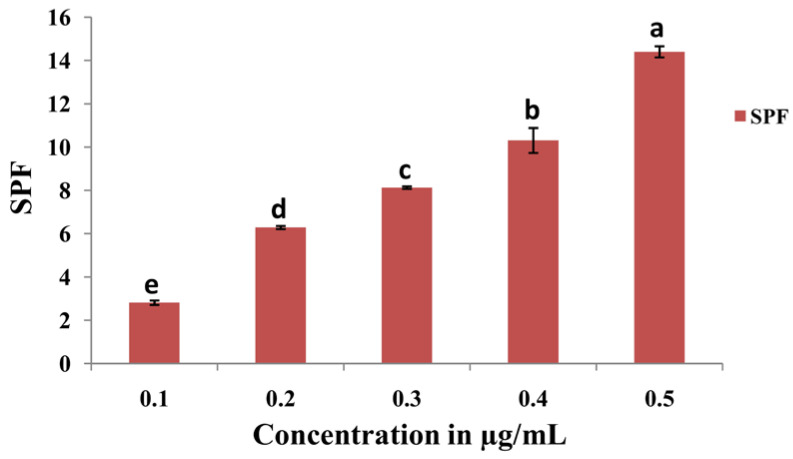
Sun protection factor of synthesized AgNPs from *Penicillium brasilianum* NP5. Data are statistically significant at *p* ≤ 0.01 by one-way ANOVA with Tukey’s B test, error bars are based on three independent experiments (means ± SD, *n* = 3). Different letters at the concentrations refer to the data that are significantly different.

**Table 1 antibiotics-12-00567-t001:** Primary screening of the isolated fungi from the collected soil samples.

Fungal Isolates	Zone of Inhibition in mm		
	*E. coli*	*S. aureus*	*C. albicans*
NP1	14	10	11
NP2	13	0	12
NP3	0	0	0
NP4	11	0	14
NP5	13	11	15
NP6	10	0	13
NP7	12	0	10
NP8	14	0	9
NP9	0	0	12
NP10	0	0	13

## Data Availability

Data available on request.

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
