# Peer review of "Myco-Nanofabrication of Silver Nanoparticles by Penicillium brasilianum NP5 and Their Antimicrobial, Photoprotective and Anticancer Effect on MDA-MB-231 Breast Cancer Cell Line"

_antibiotics, 2023, doi:10.3390/antibiotics12030567_

Round 1

Reviewer 1 Report

The paper entitled “Myco-Nanofabrication of Silver Nanoparticles by Penicillium brasilianum NP5 and their Antimicrobial, Photoprotective and Anticancer Effect on MDA MB-231 Breast Cancer Cell Line” describes the activity of fungal strain Penicillium brasilianum to fabricate AgNPs. The as-formed compound was characterized by UV-Vis, TEM, EDX, AFM, XRD, TGA, and zeta potential. Moreover, the antimicrobial activity against varied Gram-positive, and Gram-negative bacteria, and unicellular fungi was investigated. Also, anticancer activity toward breast cancer cell MDA MB-231 was checked. The manuscript contains promising data and is well-written. However, a major revision is a need before being accepted to be published in an antibiotic journal.   

Abstract

1.     The abbreviation should be mentioned completely the first time, please check and revise.

Introduction

2.     The authors should be describing the novelty of the current study because there is a lot of published literature about fungal-mediated biosynthesis of Ag-NPs and investigating their anticancer activity.

3.     The authors should be referring to recent publications related to the current point.

4.     The authors should be adding a clear hypothesis at the end of the introduction section.

Material and methods.

5.     In section “4.3”, the authors selected the most potent based on the antimicrobial activity of the crude fungal extract, but this is not logical due to the main hypothesis of the current study was to investigate the activity of fungal strain to the synthesis of silver nanoparticles. It is logical to select the most potent based on their activity to reduce silver metal to produce AgNPs according to color change and maximum SPR. Please clarify.

6.     Lines 717 and 718, “gram-negative” and “gram-positive” should be “Gram-negative” and “Gram-positive”, please revise throughout the manuscript.

7.     Line 718, “S. flexneri” please write it completely for the first time in the material and method section.

8.     Line 725, “Different concentrations” is not different concentrations, this is the different volume from the same concentration (one mg/mL).

9.     The authors investigate the antimicrobial activity for one concentration (1 mg/mL) = 1000 ppm, I think this high concentration in AgNPs. Also, it is logical to investigate different concentrations (1000 ppm, 500 ppm, 250 ppm, 125 ppm, and so on) as done in anticancer activity.

10.  The authors should be investigating the activity of fungal extract against the remaining pathogenic microbes (S. flexneri, B, cereus, and C. glabrata).

11.  There are high differences between the concentration of Ag-NPs in antimicrobial activity (1000 ppm) and anticancer activity (200 ppm, 100, ….).

12.  The biocompatibility of synthesized AgNPs against normal cell lines should be investigated. Please clarify.

13.  Please add the equation used to detect cell viability. I recommend the following to cite: https://doi.org/10.1016/j.heliyon.2020.e03943

14.  Please add the statistical analysis tool used in the current study under the title “Statistical analysis” at the end of the material and method section.

Results and discussion:

15.  Table 1 is not present in the manuscript.

16.  FTIR analysis, Can the authors detect the band represented by the AgNPs?

17.  Figure 5 (FTIR analysis) is not clear, please add another clear and sharp chart.

18.  Line 251 “surface if the AgNPs…”, please check the statement to complete.

19.  Characterization data according to AFM analysis should be mentioned in section “2.2.3”

20.  How do the authors detect PDI “polydispersity index” based on TEM analysis? I think the PDI value was extracted from DLS.

21.  Please add ±SE or ±SD beside each value in antimicrobial and anticancer activity.

22.  Data in Figure 9G (25 µg/mL, 50 µg/mL, 100 µg/mL,..) not matched with those recorded in the material and method section, subsection “4.8.1.”

23.  Please discuss the source of elements other than Ag in EDX analysis, see the following reference: https://doi.org/10.1038/s41598-022-15903-2.

24.  The conclusion should be concise to contain promising results and an overall conclusion.

Author Response

Dear Reviewers & Editor,

Thank you for your valuable comments on improvising our manuscript. The corrections have been implemented in the manuscript as instructed. The corrections made in the manuscript are highlighted. Please go through the answers for your valuable comments.

Reviewer 2 Report

Dear authors, greetings!

The manuscript “Myco-Nanofabrication of Silver Nanoparticles by Penicillium brasilianum NP5 and their Antimicrobial, Photoprotective and Anticancer Effect on MDA MB-231 Breast Cancer Cell Line” addresses the green synthesis of silver nanoparticles promoted by the extract of Penicillium brasilianum NP5 and assays the nanomaterial to investigate its antimicrobial, photoprotective and anticancer activities. However, I believe it still needs some improvements before being published in Antibiotics.

In “Abstract”, line 17 the word “isolates” and “isolated” are used very close to each other. It would be interesting to substitute one of them for a synonym. In line 30 the “In” should be written without the letter “I” being a capital one.

Regarding “Introduction”, in line 48 the first “and” can be removed. In lines 49 and 50 one of the words “various” can be substituted by a synonym. In line 55 it seems to be missing a “the” between “in” and “use”. In line 60, “actinomycetes” are not included in the term “bacteria”? In line 73 it seems to be missing a “are” between “but” and “also” and “on” should be replaced by “of”. In line 80 it seems that the idea was to write “have been reported” and not “have reported”. In line 94 the necessity for new cancer diagnostic strategies is mentioned; however, as the manuscript focuses on antimicrobial, photoprotective and anticancer activities, it is necessary to introduce a paragraph in this section to discuss the challenges regarding these fields. 

When it comes to “Results”, Figures 1A and 1B lack a scale bar. In line 124 should be written “sequence was” and not “sequence were”. In line 127 it should be written “revealed” instead of “reveals”. In line 133 it should be written “was” and not “is”. From “which” to the dot in lines 177 and 178 the text should be moved to “Discussion”. In Figure 4, it is necessary to write “(nm)” before “Wavelength” in x-axis. Regarding line 261: it is necessary to define PDI before using the abbreviation. In lines 270 and 271 was the intention to write EDX as in Figure 8 caption? In line 273 it is written that silver acts as stabilizing and capping agent; in fact fungi metabolites do that; silver ions suffer reduction from silver +1 to silver 0. It is necessary to correct that. Regarding the MTT assay, results are probably overestimated. As can be observed in Uv-vis results, the fungi extract presents absorbance at 570 nm; as can also be noted in EDX results and in methods description, it was not performed any step of washing the synthesized nanoparticles. So they still are impregnated by fungi material. As a consequence, it is necessary to repeat MTT analysis with wells dedicated to containing only NPs in the concentrations used and kit reagents; it is important to determine the amount of absorbance detected due to nanoparticles covered by extract material. It is necessary to define SPF before using the abbreviation for the first time. Figures 9, 10, 12 and 13 need to present statics results and figures’ caption, the description of what test was performed, as same as p value. 

Regarding “Discussion”, in lines 468 and 469 “isolated”, “isolates” and “isolated” are all written in a close distance; it is interesting to provide synonyms. In line 477 “confirms” needs to be substituted by “confirmed“. In line 481 it is necessary to explain the abbreviation SPR and discuss explaining the phenomenon of surface plasmon resonance in AgNPs.

When it comes to “Materials and Methods”, in section 4.2 it is necessary to add geographic coordinates of fungal collection points. In line 619 “are” should be written as “was”. In line 624 one “and” needs to be removed. It is necessary to revise the punctuation of line 626. From line 647 to line 649 dot is material for “Discussion” and should be placed there. It is necessary to remove “etc” from line 651. It is necessary to replace “is” with “was” in lines 657 and 726. Regarding MTT assay, as previously mentioned it is necessary to determine the absorbance related to nanoparticles covered by extract after exposure to kit reagents. The study lacks statistics and needs to present a section dedicated to explaining tests in this section.

Moderate English changes are required.

Author Response

(The authors gave the same response as above.)

Round 2

Reviewer 1 Report

The authors answered major issues and the manuscript is suitable for publication in the current form

Author Response

Dear Reviewer,

Thank you for your valuable comments on improvising our manuscript. Thank you for your consideration.

Reviewer 2 Report

Dear authors,

greetings!

Improvements were almost all performed successfully. However, there are still two aspects that need attention.

In Figures where statistics were added, it is necessary to add an asterisk or letter over each bar that depicts a significant result to visually indicate to readers that that result was significant.

In Figure 4, although it is not the maximum, nanoparticles present some absorbance at 570 nm as can be observed. As the authors responded: “During MTT assay the yellow colored MTT reagent reacts with mitochondrial enzyme succinate dehydrogenase which is released from cancer cells after the cell death resulting in the formation of bluish-purple formazan crystals.”. Therefore, the higher the absorbance at 570 nm at the end of MTT, the higher the cell death rate in that well. What I have pointed out is that it is necessary to repeat the MTT analysis because the presence of nanomaterial increased the absorbance at 570 nm; the value was a result of the addition: MTT assay + natural absorbance of the nanomaterial at that wavelength. The MTT needs to be repeated in the way that it needs to have wells dedicated to containing only NPs in the concentrations used. By doing that, the authors will verify the absorbance of NP to subtract from the well in which MTT was performed after exposure to that concentration of nanomaterial.  

Author Response

Dear Reviewer,

Thank you for your valuable comments on improvising our manuscript. The corrections have been implemented in the manuscript as instructed. The corrections made in the manuscript are highlighted. Please go through the answers for your valuable comments.
